# Binding of YY1/CREB to an Enhancer Region Triggers Claudin 6 Expression in *H. pylori* LPS-Stimulated AGS Cells

**DOI:** 10.3390/ijms241813974

**Published:** 2023-09-12

**Authors:** Jorge H. Romero-Estrada, Luis F. Montaño, Erika P. Rendón-Huerta

**Affiliations:** Laboratorio de Inmunobiología, Departamento de Biología Celular y Tisular, Facultad de Medicina, Ciudad Universitaria, Ciudad de México 04510, Mexico; jorge.hieratic@gmail.com

**Keywords:** claudin 6, gastric cancer, transcriptional regulation

## Abstract

Aberrant expression of the tight junction protein claudin 6 (CLDN6) is a hallmark of gastric cancer progression. Its expression is regulated by the cAMP response element-binding protein (CREB). In gastric cancer induced by *Helicobacter pylori* (*H. pylori*) there is no information regarding what transcription factors induce/upregulate the expression of CLDN6. We aimed to identify whether CREB and Yin Yang1 (YY1) regulate the expression of *CLDN6* and the site where they bind to the promoter sequence. Bioinformatics analysis, *H. pylori* lipopolysaccharide (LPS), YY1 and CREB silencing, Western blot, luciferase assays, and chromatin immunoprecipitation experiments were performed using the stomach gastric adenocarcinoma cell line AGS. A gen reporter assay suggested that the initial 2000 bp contains the regulatory sequence associated with *CLDN6* transcription; the luciferase assay demonstrated three different regions with transcriptional activity, but the −901 to −1421 bp region displayed the maximal transcriptional activity in response to LPS. Fragment 1279–1421 showed CREB and, surprisingly, YY1 occupancy. Sequential Chromatin Immunoprecipitation (ChIP) experiments confirmed that YY1 and CREB interact in the 1279–1421 region. Our results suggest that CLDN6 expression is regulated by the binding of YY1 and CREB in the 901–1421 enhancer, in which a non-described interaction of YY1 with CREB was established in the 1279–1421 region.

## 1. Introduction

Epithelial tight junctions are dynamic structures formed via the aggregation of several different transmembrane proteins (Zonula occludens-1, occludins, junctional adhesion molecules, Marvel3, and claudins) [1] that determine the barrier properties between the plasma membranes of adjacent cells [2]. Claudins are a family of 27 different proteins [3] whose homo- and heterodimeric interactions form tight junction strands in a tissue-specific combination and exert pore-forming activity [4,5]. CLDN6 expression is developmentally regulated in embryonic epithelia [6], mainly the fetal stomach, lung, and kidney [7], and rarely in healthy adult tissues [8]. Deregulation of CLDN6 expression and distribution has been associated with epithelial cancer progression [9] in non-small-cell lung, ovarian, cervical, and breast carcinomas [10,11,12,13], but in the latter, its function as a tumor-promoting or tumor suppressor gene has been recognized [14,15]. Similarly, abnormal expression of CLDN6 has been widely recognized in gastric cancer [16,17], in which its expression has been clearly associated with enhanced invasiveness and metastatic properties [18,19] via well-defined transcription factors [20]. Extended exposure to *H. pylori* LPS increases the expression of CLDN6 in AGS cells [21].

Mongolian gerbils infected with *H. pylori* show up-regulation and activation of CREB that correlate with early cellular inflammation and ulceration [22]. In silico studies reveal that CREB-mediated transcription regulates diverse cellular responses, such as cell proliferation and apoptosis [23], and that *claudin 6* expression is regulated by CREB, amongst other transcription factors [24]. Interestingly, chronic inflammation-associated IL1b signaling regulates the expression and activation of CREB through an extracellular signal-regulated kinase 1, 2 (ERK 1/2) dependent mechanisms [25], a pathway closely related to CLDN6 expression [21].

Diverse nuclear processes, including enhancer–promoter interactions, are regulated, among others, by CCCTC-binding factor (CTCF) [26]. YY1 contributes to enhancer–promoter structural interactions; it possesses one activation and two repression domains [27]. It binds to the CGCCATNTT sequence located in many different promoters and acts as a regulator of enhancer–promoter loops analogous to DNA interactions mediated by CTCF [28]. Monomeric YY1 bound to DNA is capable of dimerizing, forming DNA loops interacting directly with DNA sequences or through other proteins [29]. YY1 is a sequence-specific DNA multi-domain binding transcription factor that activates or represses genes during cell growth and differentiation [30,31], and it has a role in the control of epithelial mesenchymal transition (EMT) [32]. It interacts with CREB-binding protein [33] and histone deacetylase-1, -2, and -3 [27]. YY1 plays a vital biological role in the remodeling and regulation of angiogenesis, tumor metabolism, and immunity in the microenvironment of a great variety of tumors, including gastric cancer [34].

The interaction between YY1 and CREB has been established in viral infections [33,35] and gastric tumors [36]. But for *H. pylori*-induced gastric cancer, there is no information regarding what transcription factors induce/upregulate the expression of CLDN6. The aim of this paper is to identify whether CREB and YY1 transcription factors regulate the expression of CLDN6 and to identify the site where they bind to the promoter sequence.

## 2. Results

### 2.1. Bioinformatic Analysis

As YY1 interacts with CREB, a Transcription Factor Binding (TFBind) analysis was performed to determine all the possible YY1 and CREB binding sites in a promoter region close to the location where *CLDN6* transcription is initiated; because the number of possible binding sites was extensive, we selected sites with a minimal similitude value of 0.8 for YY1 and CREB. Nine YY1 binding sites were found, but one with a similitude value of 0.9 was in a site very close to where transcription is initiated. Concerning CREB, 17 binding sites with a similitude value between 0.80 and 0.89, and three with a similitude value greater than 0.9, were selected. Interestingly, a luciferase assay determined that none of the 0.9-value YY1 or CREB participated in the transcriptional activity after LPS stimulation (Figure 1—YY1 and CREB binding sites in *CLDN6* promoter).

### 2.2. H. pylori LPS and YY1/CREB Expression

CLDN6 expression is enhanced by *H. pylori* LPS [21]; nevertheless, the effect of LPS upon the expression of YY1 and CREB has not been studied. Our results confirmed that LPS exposure initiated CLDN6 expression after 12 h, reaching its peak value after 48 h (*p* < 0.05). Interestingly, this increase was apparently subordinated to a significant up-regulation (*p* < 0.01) of YY1 and CREB transcription factors that initiated 4 h after LPS exposure and reached its maximum value after 12 and 24 h (Figure 2A,B), respectively.

To determine the region that regulates CLDN6 expression, we performed a gen reporter assay using the results of the bioinformatics analysis, which suggested that the initial 2000 bp contained the regulatory sequence associated with CLDN6 transcription.

### 2.3. Cloning of CLDN6 Promoter

To evaluate the transcriptional activity of the *CLDN6* promoter, three different regions were cloned in the pMetLuc2 (−2000, −1421, −901 bp) (Figure 3A). Transfected AGS cells showed, as determined via the luciferase assay, that the three different regions displayed transcriptional activity that was independent of LPS presence (Figure 3B). The apparent difference in activation with and without LPS in the results of the −901 region did not reach a statistically significant difference. It is worth noticing the enhanced activation of the −1421 region in 48 h LPS-stimulated cells. To compare the responses of the different regions adequately, normalized data were used to demonstrate that the −901 to −1421 region displayed the maximal transcriptional activity in response to LPS (Figure 3C). The difference started to be significant from 12 h and reached a peak after 24 h that maintained a plateau behavior after 48 h. Not surprisingly, the complete −2000 bp region behaved almost identically to the −1421 region from 24 h, thus suggesting that the transcriptional activity of the −901 region is not dependent on LPS stimulation.

### 2.4. Regulation of CLDN6 Expression

To determine the role of YY1 and CREB in *CLDN6* transcription regulation, silencing experiments using specific siRNAs were done. Figure 4A represents the silencing control experiments where efficient silencing of both transcription factors was achieved. The results showed that knocking down YY1 or CREB diminished the expression of CLDN6, thus emphasizing the importance of YY1 transcription factor in the regulation of CLDN6 expression. The results also demonstrated that YY1 has a regulatory role in CREB expression, suggesting a highly relevant function for this protein in gastric cancer. CREB silencing in LPS-treated cells significantly diminished the expression of YY1 (Figure 4B,C), suggesting that LPS triggers the establishment of a loop constituted by the CREB and YY1 interaction. These results were independent of LPS stimulation, and the differences against control experiments were highly significant (*p* < 0.0001).

### 2.5. Interaction of YY1/CREB in CLDN6 Promoter

As the luciferase analysis had already shown, the 901–1421 region possessed the elements that control CLDN6 expression in LPS-treated cells. The chromatin immunoprecipitation assays showed YY1 and CREB occupancy sites in the 901–1018 bp fragment; the 1018–1149 fragment only had YY1 occupancy, and the 1149–1279 fragment did not show occupancy signals for either transcription factor; these sites corresponded to those predicted via the bioinformatic analysis. The 1279–1421 fragment showed CREB occupancy as predicted, but surprisingly, YY1 occupancy was also demonstrated despite not being predicted via the bioinformatic analysis (Figure 5).

Because of the surprising presence of a YY1 occupancy site in 1279–1421, we looked for possible YY1 binding sites with a minimal similitude value of 0.5 onwards in this region; the analysis did not show a putative binding site. Because ample bibliographic information validates the spatial interaction of both proteins, we performed sequential ChIP experiments in all three regions to determine whether the presence of YY1 in the 1279–1421 region is secondary to its interaction with CREB. The results confirmed, as expected, that YY1 and CREB interact in the 1279–1421 region, thus confirming our hypothesis (Figure 6).

## 3. Discussion

Claudins, a large family of transmembrane structural proteins, are fundamental to tight junction function and regulation linked to associated regulatory and scaffolding proteins [37]. They exhibit cell-type-specific and tissue-specific expression patterns [38]. Still, post-translational modifications can alter tight junction protein binding events and barrier function [39]. The fetal stomach tissue expresses CLDN6, which is developmentally regulated in mouse embryonic epithelia and is one of the earliest proteins expressed in embryonic stem cells committed to the epithelial fate [6,7]. CLDN6 expression in breast cancer is regulated by transcription factors such as HIF-1a, FoxA2, Gata6, and TTF-1 [40,41] at the *claudin 6* promoter [42]. CLDN6 expression is significantly upregulated in different types of cancer, including gastric cancer [17,43,44]. CLDN6 overexpression in the gastric adenocarcinoma cell line AGS has proved to play a significant role in cell proliferation, migration, and invasion [18,19], but very little is known about the transcription factors that regulate CLDN6 expression.

A previous analysis of claudin expression in the AGS cell line suggested that the transcription factors CREB and YY1 could be regulators of *CLDN6*. The bioinformatics analysis results showed that CREB and YY1 have 17 and 9, respectively, probable binding sites 2000 bp upstream of the site where transcription of *CLDN6* is initiated. Our results confirmed that *H. pylori* LPS induced, in the context of a chronic inflammatory process, the expression of CLDN6 in the AGS cell line [21]. Still, it also induced the expression of CREB and YY1 protein. The results of a Pearson correlation analysis between these transcription factors and CLDN6 expression suggested that both proteins, YY1 and CREB, could be associated with CLDN6 expression.

YY1 is a zinc finger protein that can activate or repress transcription, depending on its interactions with other transcription factors [45,46,47], such as the nuclear protein CBP, which is a coactivator for the transcription factor CREB [48]. CREB is strongly associated with inflammation and progression genes [49], and YY1, which is upregulated in the AGS cell line, is associated with gastric cancer progression [50]. Nevertheless, there is no information regarding the DNA region where these transcription factors bind.

Bacterial LPS regulates gene transcription by binding YY1 complexes to the CCAAT enhancer binding protein 1 (C/EBP1) or long non-coding RNA (lncRNA) via inflammatory cytokines [51,52]. Our results showed that the presence of CREB is indispensable for its interaction with YY1. The triggering mechanism in our model is highly likely to be mediated by pro-inflammatory cytokines induced by *H. pylori* LPS. lL-1b and TNFα regulate gene transcription via the C/EBP1, C/EBP2, and YY1 elements in epithelial cells [53]. YY1 function is regulated through the IL-4/STAT6 signaling pathway in tumor-associated immune cells [54]. LPS is not the only pathogenic factor in *H. pylori* capable of inducing gastric carcinogenesis; cytotoxin-associated gene A (CagA) protein, another major and substantial component of *H. pylori*, is known to provoke genomic hypermutations, DNA double strand breaks, subverted DNA damage responses that include the down-regulation of DNA repair genes, and transcriptomic and proteomic alterations that increase the chance of generating a “hit-and-run mechanism” for gastric carcinogenesis [55,56,57].

To identify the specific regulatory regions of the 2000 bp promoter regions where CREB and YY1 binding sites were found, the complete promoter region and two fragments were cloned in the pMetLuc2 vector and evaluated via a luciferase activity assay. The results suggested that two CREB sites and two YY1 sites in the DNA fragment corresponding to the 901 bp–1421 bp region are significantly important for regulating the *CLDN6*. The ChIP assays confirmed the results and revealed a YY1 binding site in the 1279–1421 bp region not predicted via the bioinformatics analysis. The distance between these CREB and YY1 sites was identified, suggesting that the 1279–1421 region is an enhancer [58]. Gene expression is a precisely controlled process where enhancers function in a tissue-specific manner [59]. Enhancers in higher eukaryotes are physically separated along the genome from the target gene promoters, and there are three possibilities for enhancer–promoter communication: tracking, linking, and looping [60]. YY1 is known to interact with DNA in a monomeric or a dimeric manner through not only its zinc fingers but also bridging with other transcription co-factors [28]. Our results strongly suggest that the binding of YY1 with CREB mediates the formation of a “loop” and, thus, initiates *CLDN6* transcription, but the precise mechanism remains to be solved. YY1 has been recognized as a structural regulator of enhancer–promoter loops [28]. YY1 and CREB binding has been confirmed in the fourth exon of lymphotoxin-b in chromosome 6p21 in Jurkat T cells [61].

The mechanism through which YY1 functions both as a transcriptional activator and repressor depends in its acetylation/deacetylation [27]. Still, it is likely that the proteins and sequence-specific DNA-binding transcriptional activators and coregulatory molecules, such as CREB, with which YY1 interact determine its function [29,62]. Our results suggest that the interaction between YY1 with CREB in AGS cells stimulated with *H. pylori* LPS is involved in the transcriptional initiation and activation of *CLDN6*, a perfectly defined tight junction protein associated with enhanced gastric cancer progression and invasiveness [19].

Our results suggest that CLDN6 expression, a protein associated with gastric epithelial cancer progression, is regulated by the binding of YY1 and CREB in the 901–1421 enhancer in which a non-described interaction of YY1 with CREB was established in the 1279–1421 region.

## 4. Materials and Methods

### 4.1. Reagents

Dulbecco’s modified Eagle’s medium (DMEM), fetal bovine serum (FBS), L-glutamine, sodium pyruvate, insulin, Dulbecco’s phosphate-buffered saline (PBS), streptomycin–penicillin, bovine albumin, and total antibody compensation beads were from Invitrogen (Life Technologies Corp, Carlsbad, CA, USA). Monoclonal anti-claudin-6 (sc-393671), anti-YY1 (sc-7341), anti-GAPDH (sc-47724), anti-actin (sc-32251), and goat anti-mouse IgG were from Santa Cruz Biotechnology (Dallas, TX, USA). Monoclonal anti-CREB (MA1-083) and Lipofectamine 2000 (11668027) were from ThermoFisher Scientific (Waltham, MA, USA). The Molecular Biology Kit, EZ-10 Spin Column Plasmid DNA Miniprep Kit was from Southern Labware (Cumming, GA, USA). The ready-to-glow secreted luciferase reporter assay (Cat # 631727) and pMetLuc2 reporter vector (Cat # 631729) were from Takara Bio USA (San Jose, CA, USA). Primers were designed using TFBIND [63] and were synthetized by Integrated DNA Technologies (Coralville, IA, USA). T4 DNA ligase (cat #M0202S) was from New England BioLabs (Ipswich, MA, USA). The chromatin immunoprecipitation EZ-ChIP (cat #17-371) was from Merck KGaA (Darmstadt, Germany). Ruby Hot Start Master (2×) (cat # PCR-165L) was from Jena Bioscience GmbH (Jena, Germany).

### 4.2. Cell Culture

Human gastric adenocarcinoma (AGS) (CRL-1739, American Type Culture Collection Manassas, Manassas, VA, USA) cells (1 × 10^6^) were cultured in sterile P-100 Petri dishes with DMEM supplemented with 5% FBS, 0.1 U/mL of insulin, a 1% streptomycin–penicillin solution, 2 mmol/L of L-glutamine, and a 2 mmol sodium pyruvate solution at 37 °C in a humid environment containing 5% CO_2,_ until reaching >90% confluence. Luciferase experiments were performed with 8 × 10^5^ cells/well seeded in 6 well Tissue Culture Plates (Biocompare, San Francisco, CA, USA) for 48 h until they reached 90% confluence, which represents 1 × 10^6^ cells/well. The LPS treatment of AGS cells was performed in p-100 culture dishes once they reached 80–90% confluence. All the experiments were immediately performed with these confluent AGS cell cultures in their third passage.

### 4.3. Bioinformatic Analysis

The search for sequences with potential transcriptional relevance in the present study was performed using TFBIND (https://tfbind.hgc.jp/, accessed on October 2019). We used a 0.8 cut-off instead of the 0.5 value for both transcription factors.

### 4.4. YY1 and CREB Silencing

Specific YY1 and CREB small interfering RNAs (Santa Cruz Biotechnology, siRNAYY1 sc-36864, siRNACREB sc-35111) were used to transfect AGS cells. 3 × 10^5^ cells were seeded in 6 well culture plates with 10% SFB supplemented antibiotic-free DMEM for 40 h at 37 °C in a 5% CO_2_ atmosphere until an 80% confluence was reached. Cells were transfected with the YY1 or CREB siRNA, following the manufacturer’s protocol, and cultured in transfection media for a 6 h period at 37 °C in a 5% CO_2_ atmosphere, followed by 20 h incubation in DMEM. At the end, culture media were eliminated from the wells, and 2 mL of 1% streptomycin–penicillin solution and 10% FBS supplemented DMEM were added per well to the transfected cells, which were incubated for 24 h at 37 °C in a humid 5% CO_2_ atmosphere before extracting total cell protein. A Western blot analysis was performed to corroborate the silencing of YY1 and CREB at the protein level.

### 4.5. Preparation of LPS Helicobacter pylori and Exposure to AGS Cells

*Helicobacter pylori* (strain J99) LPS were kindly donated by Dr. Victor R. Coria, Instituto Nacional de Pediatría, México. AGS cells were treated with 10 ng/mL of *LPS H. pylori* for 4 h, 8 h, 12 h, 24 h, and 48 h.

### 4.6. Protein Extraction

Control, LPS-treated, and transcription factors silenced AGS cells were washed twice with PBS and scraped with 1 mL of an ice-cold lysis buffer (150 mM NaCl/50 mM Tris/1 mM EGTA/1 mM EDTA/1% IGEPAL/0.1% Sodium deoxycholate/0.1% SDS + protease and phosphatases inhibitors). Cell suspensions were sonicated for 1 min at 25% amplitude, followed by centrifugation for 30 min at 17,000× *g* (4 °C). The supernatants were recovered, and a Bio-Rad Protein Assay determined the total protein concentration (Cat. # 500-0009, Bio-Rad, Hercules, CA, USA).

### 4.7. Western Blot

For Western blot analysis, 20 µg of protein was resolved on 13% SDS/PAGE and transferred to nitrocellulose membranes in a Bio-Rad semi-dry blotting system for 1 h at 120 mA. Membranes were blocked with 5% non-fat dry milk in Tris-buffered saline (TBS) for 1 h, washed twice with TBS, and incubated with the relevant primary antibody diluted in Tween 20/TBS (TTBS) overnight at 4 °C (anti claudin-6 (1:200), anti-YY1 (1:400), anti-CREB (1:500), anti-GAPDH (1:5000), and anti α-actin (1:500)). Membranes were washed with TTBS and incubated with horseradish peroxidase-labeled secondary goat anti-mouse IgG (1:4000) diluted in TTBS for 1 h at room temperature. Afterward, membranes were washed thrice with TTBS and once with TBS before antibody binding was detected via chemiluminescence using the Supersignal West Dura Extended Duration Substrate (Thermo Fisher Scientific, Waltham, MA, USA) as a substrate. Equal protein loading was confirmed in all the experiments by determining β-actin as a loading control. All experiments were done in triplicate. The quantitative analysis of the Western blot bands was performed using ImageJ v 1.53 software, which calculates the intensity of the gel band, measuring the amount of pixels/sq.in. The values were determined for each band. The results are expressed as dots per point (DPP) and intensity.

### 4.8. Plasmid Construction and Luciferase Assay

The −2000, −1421, and −901 regions of the *claudin 6* promoter were amplified from AGS genomic DNA using the following primers: −2000 forward: *5′CCAGCCGGTGATCTAGTCC3′*; −1421 forward: *5′GCCACTACAGCTTTGTTAAGGG3′*; and 901 forward: *5′CGGGCACCTGTAGTAGTCC3′*. We used the same reverse primer: *5′AATTCCTAGGCCGAGTGTCG3′*. The amplified DNA was cloned in pMetLuc2 plasmid using DNA ligase T4. The result of this ligation was used to transform competent *E. coli* DH5α to get transformed colonies. Plasmid DNA was extracted using the Southern Labware EZ-10 Molecular Biology Kit, according to the manufacturer´s instructions. The purified DNA was used to transfect AGS cells using lipofectamine 2000; these transfected AGS cells were treated with *H. pylori* LPS 10 ng/mL for 4, 8, 12, 24, and 48 h. Luciferase activity was measured using the Ready-to-Glow secreted luciferase reporter assay (Clontech Lab, Mountain View, CA, USA) in a Perkin Elmer Wallac 1420 Victor Spectrophotometer. The values obtained from these experiments were normalized according to Schagat T. of Promega Corporation (Cell Notes Issue 17, 2007).

### 4.9. Chromatin Immunoprecipitation

A ChIP assay was performed using Millipore EZ-ChIP kit (Merck KGaA, Darmstadt, Germany) according to the manufacturer’s instructions. In brief, AGS cells were treated with *H. pylori* LPS 10 ng/mL for 24 h before being fixed with 3.7% formaldehyde, lysed, and sonicated, as previously mentioned. Immunoprecipitation was then performed with CREB and YY1 antibodies, and purified DNA fragments were analyzed via qPCR, using Ruby Hot Start Master (2×) according to the manufacturer’s instructions. The following primers were used: 901–1018 forward: *5′GCACTACAGCTTTGTTAAGGG3′*; reverse: *5′CACTACCACGCCCGGCTAAC3′*; 1018–1149 forward: GTTAGCCGGGCGTGGTAGTGGGC; reverse: CGCCCAGTCTGGAGTGCAATGG; 1149–1279 forward: CCATTGCACTCCAGACTGGGCG; reverse: TTTCCTGACCTCGTGATCTGCCC; and 1279–1421 forward: GGGCAGATCACGAGGTCAGGAAA; reverse: GGGACTACTACAGGTGCCCG. The results of these assays were analyzed in a 3.5% agarose gel, and the quantitative analysis of the bands’ images was performed using ImageJ v 1.53 software, which calculates the intensity of the gel band, measuring the amount of pixels/sq.in. The values were determined for each band. The results are expressed as dots per point (DPP) and intensity.

### 4.10. Statistical Analysis

Statistical analyses were performed using the GraphPad Prism software, version 4 (GraphPad Software, Boston, MA, USA). All values are expressed as means ± standard deviations (SDs). Statistical significance in a one-way analysis of variance (ANOVA), followed by a post hoc Dunett’s test and selected pairs comparison test, was set to *p* < 0.05 (*), *p* < 0.01 (**), *p* < 0.001 (***) or *p* < 0.0001 (****) versus the control condition, and “*n*” represents the number of independent experiments.

## Figures and Tables

**Figure 1 ijms-24-13974-f001:**
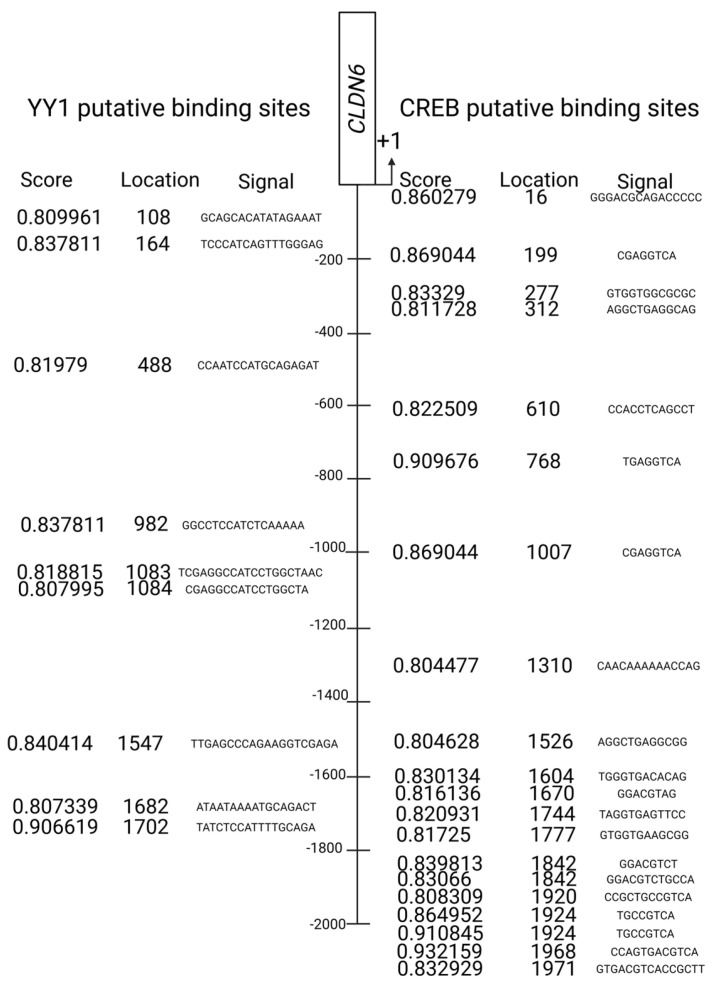
TFBind software (https://tfbind.hgc.jp/) was used to select putative binding sites for YY1 and CREB transcription factors based on a minimal similitude score of 0.8. Location refers to the position of the initial nucleotide in the identified sequence. Signal lists the sequence in the *CLDN6* promoter where there is a high probability that the transcription factors bind.

**Figure 2 ijms-24-13974-f002:**
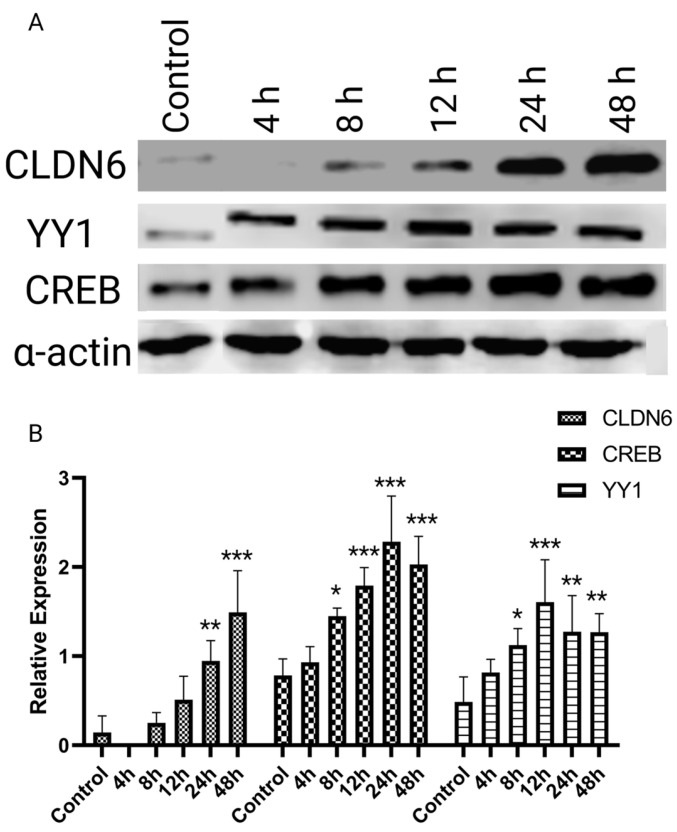
Expression of CLDN6, YY1, and CREB in *H. pylori* LPS-stimulated AGS cells. (**A**) AGS cells were exposed to 10 ng/mL of *H. pylori* LPS, and the expression of CLDN6, YY1, and CREB was evaluated at different time intervals, (**B**) Quantitative representation of the densitometric evaluation of the western blot shown in A. All the experiments were performed in triplicate. The difference in expression was determined using ANOVA and Dunnett’s test. *, **, and *** represent 0.05, 0.01, and 0.001 *p*-values, respectively.

**Figure 3 ijms-24-13974-f003:**
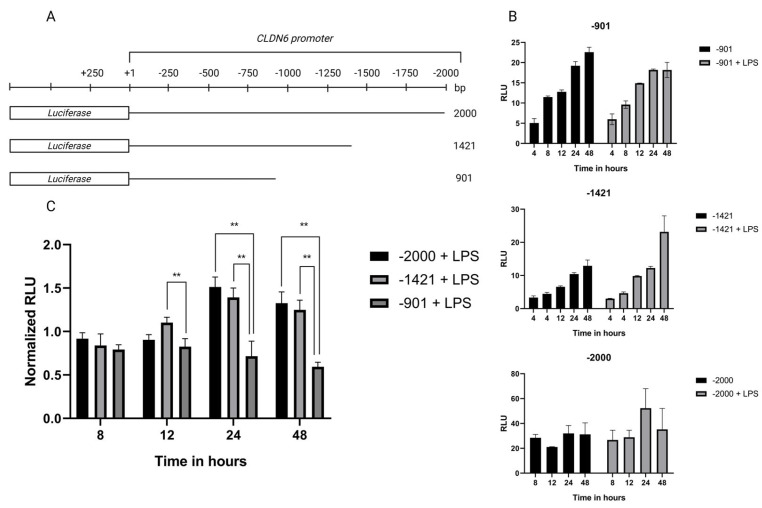
Transcriptional activity of different regions of the *CLDN6* promoter. AGS cells were exposed to 10 ng/mL of *H. pylori* LPS, and the transcriptional activity of the *CLDN6* promoter was determined in (**A**) different regions cloned in the pMetLuc2 vector and (**B**) transcriptional activity in relative units (RLU) in the cloned regions (the values did not reach statistical significance). (**C**) The normalized RLU data of the results showed a significant difference in transcriptional activity between the −901 and the −1421 region from 12 h of LPS exposure. All the experiments were performed in triplicate. The difference in expression was determined using ANOVA and Dunnett’s test. ** represent 0.01 *p*-value.

**Figure 4 ijms-24-13974-f004:**
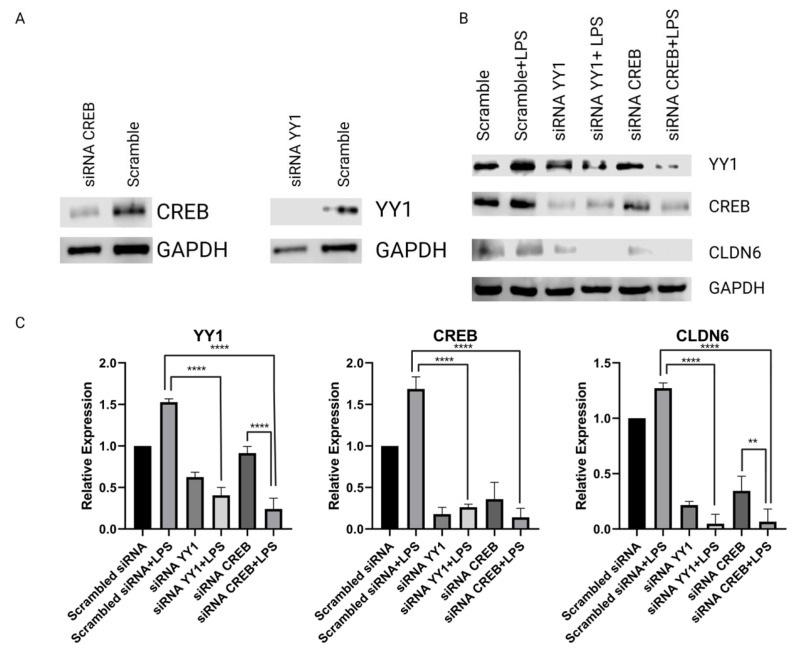
YY1 and CREB silencing in *H. pylori* LPS-treated AGS cells. Specific siRNA for YY1 and CREB were used to silence their expression in *H. pylori* LPS-treated AGS cells. (**A**) CREB and YY1 silencing; scramble siRNA was used as a negative control. (**B**) Effect of YY1 and CREB silencing on CLDN6 expression. (**C**) Quantitative expression of the results shown as histograms. All the experiments were performed in triplicate. All the experiments were performed in triplicate. The difference in expression was determined using ANOVA and Dunnett’s test. ** and **** represent 0.01 and 0.0001 *p*-values, respectively.

**Figure 5 ijms-24-13974-f005:**
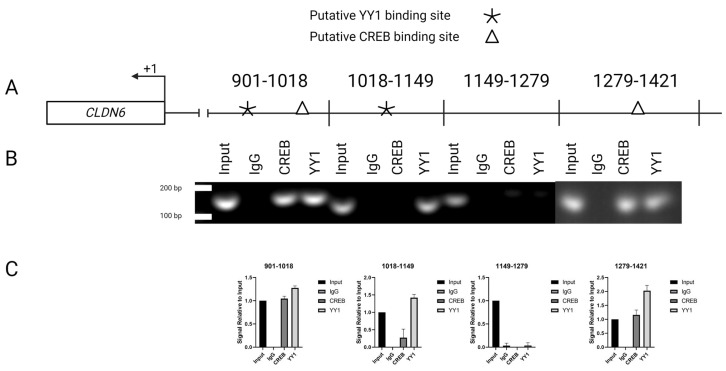
Interaction of YY1 and CREB in the *CLDN6* promoter. (**A**) The putative binding sites of YY1 and CREB in the 901–1421 region; (**B**) CHIP assay corroborating the occupancy sites and the presence of a non-predicted YY1 site in the 1279–1421 region; (**C**) histograms presenting the relative signal quantification compared to the control (input). All the experiments were performed in triplicate.

**Figure 6 ijms-24-13974-f006:**
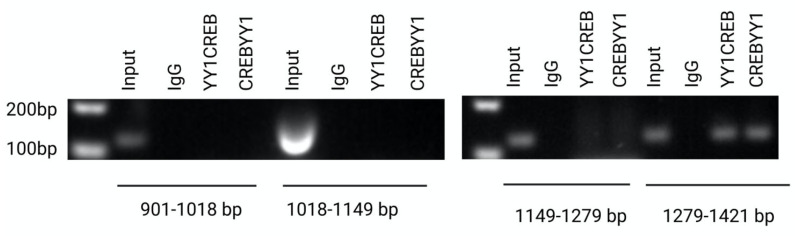
Sequential CHIP assays. Confirmation of YY1 and CREB interaction in the 1279–1421 region via sequential CHIP assays. All the experiments were performed in triplicate.

## Data Availability

Data are available upon reasonable request.

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
