# Peer review of "Binding of YY1/CREB to an Enhancer Region Triggers Claudin 6 Expression in H. pylori LPS-Stimulated AGS Cells"

_ijms, 2023, doi:10.3390/ijms241813974_

Round 1

Reviewer 1 Report

1. This is a straight forward biding characterization of YY1 and CREB2 in promoter of Claudin 6.

2. This is a single cell line study and discussion on HP gastric carcinogenesis would be overmatching, but many other papers do. Line 272 - 275.

3. Line 263 (ref) should be filled with the correct and updated references.

4. Line 116, 118: heading: cloning of caludin 6 promoter should be expressed in large scale like Line 132 because it is human gene.

5. HP carcinogenesis has several theories. Provide the basic background of HP LPS. Many readers would be more familiar with cagA product in the context HP induced changes of signal transduction (Hatakeyama group)

6. A graphical abstract would be helpful especially on the loop of two binding protein.

Author Response

To reviewer

I have carefully read your comments and have made all the suggested modifications

Reviewer 1

Q 1 and Q2. Both comments are deeply appreciated. I would like to mention that I understand we performed all our experiments with only one cell line, but our aim was not to compare cell lines but determine if YY1 and CREB were present in the 901-1420 region of the CLDN6 promoter as the bioinformatic analysis that we performed suggested.

Q 3. Line 263 (ref) should be filled with the correct and updated references.

Answer. An adequate reference has been properly included in the text and the reference listing.

Q 4. Line 116, 118: heading: cloning of caludin 6 promoter should be expressed in large scale like Line 132 because it is human gene.

Answer. We have thoroughly checked the text so when the text refers to the human protein, we have written CLDN6 in capital letters whereas when the text refers to the gen, we have written CLDN6 in capital letters and italics.

  1. HP carcinogenesis has several theories. Provide the basic background of HP LPS. Many readers would be more familiar with cagA product in the context HP induced changes of signal transduction (Hatakeyama group).

Answer. Gastric carcinogenesis is a complex process triggered by numerous factors. Nevertheless, we agree with the comment and therefore we did modify the relevant paragraph in the discussion section emphasizing the role of CagA participation in the carcinogenesis process, extensively reviewed in Hatakeyama group papers which we have included in the reference list.

  1. A graphical abstract would be helpful especially on the loop of two binding protein.

Answer. We believe that our findings are clearly represented in the graphical abstract that we submitted, nevertheless, and considering the reviewer comment, we have enlarged the loop components in the diagram.

Reviewer 2 Report

This manuscript aims to study the regulatory mechanism of CLDN6 expression via the binding of YY1/CREB in H.pylori LPS-stimulated AGS cells. 

The overall design is sound and the topic is of interest to the field because CLDN6 has been shown to be significantly associated with gastric cancer progression and tumor invasiveness. 

1. In Figure2, it was described that LPS exposure increased the expression of CLDN6 and reached the peak at 12 hours. However, it is not shown in the figure as the expression for 24h and 48h is higher than 12h. Please address this or clarify the description. 

2. For Figure3 and Figure4, the authors should perform Mann-Whitney U test to compare the expression levels between two groups. Statistical significance is not shown in Figure4. 

3. Have the authors performed Chip-seq of the AGS cells? If so, did the authors compute the peaks in the enhancer region for CLDN6? It will be interesting to see some computational results from the Chip-seq experiments. 

Author Response

To Reviewer 2

I have carefully read your comments and have made all the suggested modifications

  1. In Figure2, it was described that LPS exposure increased the expression of CLDN6 and reached the peak at 12 hours. However, it is not shown in the figure as the expression for 24h and 48h is higher than 12h. Please address this or clarify the description. 

Answer. We never mentioned that CLDN6 expression reached the peak value at 12 hs but it is probable that the way the paragraph was written might have been confusing therefore we have rewritten the paragraph and we believe it is much more precise.

  1. For Figure3 and Figure4, the authors should perform Mann-Whitney U test to compare the expression levels between two groups. Statistical significance is not shown in Figure4. 

Answer. In relation to the comment related to the statistical method used in figures 3 and 4, we decided not to modify our analysis because the parameters that define which methods should be used in different experimental conditions and, as pointed in Mishra P et al., in Ann Card Anaesth 2019, the one-way ANOVA test is the parametric choice for three or more unpaired groups. The reason behind our selection of the one-way ANOVA was based on the consideration that the behavior in biological processes tend to show a normal distribution as determined by Q-Q plots (Vetter TR, Anesth Anal 2017). The analysis of the theoretical vs the sampled quantile performed value using PRISMA software showed a normal distribution of the data (shown in the following figures)

Figures footnote. YY1, Claudin 6 and CREB, respectively.

Due to the normal distribution of the data we are certain that the application of the Mann-Whitney U test, a non-parametric test, is not convenient

  1. Have the authors performed Chip-seq of the AGS cells? If so, did the authors compute the peaks in the enhancer region for CLDN6? It will be interesting to see some computational results from the Chip-seq experiments. 

Answer. Because our research aim was to determine if YY1 and CREB were present in the 901-1420 region of the CLDN6 promoter we decided to focus on the above-mentioned region. ChIP-seq is certainly a powerful tool but was not required for our purpose.
